# Growth tradeoffs produce complex microbial communities on a single limiting resource

Michael Manhart [1] & Eugene I. Shakhnovich [1]

The relationship between the dynamics of a community and its constituent pairwise interactions is a fundamental problem in ecology. Higher-order ecological effects beyond pairwise interactions may be key to complex ecosystems, but mechanisms to produce these effects remain poorly understood. Here we model microbial growth and competition to show that higher-order effects can arise from variation in multiple microbial growth traits, such as lag times and growth rates, on a single limiting resource with no other interactions. These effects produce a range of ecological phenomena: an unlimited number of strains can exhibit multistability and neutral coexistence, potentially with a single keystone strain; strains that coexist in pairs do not coexist all together; and a strain that wins all pairwise competitions can go extinct in a mixed competition. Since variation in multiple growth traits is ubiquitous in microbial populations, our results indicate these higher-order effects may also be widespread, especially in laboratory ecology and evolution experiments.

[1] Department of Chemistry and Chemical Biology, Harvard University, 12 Oxford Street, Cambridge, MA 02138, USA. Correspondence and requests for materials should be addressed to M.M. (email: mmanhart@fas.harvard.edu)

Complex communities with a large number of distinct species or strains abound in both nature[1,2] and the laboratory[3,4]. A fundamental problem in ecology is to understand the relationship between a community's behavior and the pairwise interactions of its constituents[5–8]. In particular, the key question is to what extent these pairwise interactions determine the behavior of the community as a whole. However, ecologists have long considered the possibility of higher-order effects such that the interaction between pairs of strains can be altered by the presence of additional strains[7–11]. These higher-order effects may cause a community to be fundamentally different than the sum of its pairwise interactions and can play an important role in stabilizing coexisting communities[12,13]. Although these higher-order effects may be essential to accurately predict the ecological and evolutionary dynamics of a population, their underlying mechanisms remain poorly characterized.

The relative simplicity and experimental tractability of microbes make them convenient for studying this problem. Most well-known ecological effects in microbes are mediated by cross-feeding interactions or the consumption of multiple resources[14]. For example, long-term coexistence of distinct strains is often believed to depend on the existence of at least as many resource types as coexisting strains, according to the principle of competitive exclusion[15,16]. However, theoretical and experimental work has demonstrated that tradeoffs in life-history traits alone—for example, growing quickly at low concentration of a resource versus growing quickly at high concentration, but with only a single resource type and no other interactions—are sufficient to produce not only stable coexistence of two strains[17–20] but also non-transitive selection[21], in which pairwise competitions of strains form a rock-paper-scissors game[22].

Variation in multiple growth traits, such as lag time, exponential growth rate, and yield (resource efficiency), is pervasive in microbial populations[23–25]. Not only are single mutations known to be pleiotropic with respect to these traits[26,27], but even genetically-identical lineages may demonstrate significant variation[28,29]. The ecological effects of such variation, however, are unknown in large populations with many distinct strains simultaneously competing, as is generally the case for microbes.

Here we study a model that shows how covariation in growth traits can produce complex microbial communities without any interactions among cells beyond competition for a single limiting resource. We focus on variation in lag times, exponential growth rates, and yields since they are the traits most easily measured by growth curves of individual strains[30]. We show that covariation in these traits creates higher-order effects such that the magnitude and even the sign of the selection coefficient between a pair of strains may be changed by the presence of a third strain. These higher-order effects can produce nontrivial ecological phenomena: an unlimited number of strains can form a multistable community or neutrally coexist, potentially with a single keystone strain stabilizing the community[31,32]; strains that coexist in pairs do not coexist in a community all together; and a strain that wins all pairwise competitions can go extinct in a mixed competition. Our model can be combined with high-throughput measurements of microbial growth traits to make more accurate predictions of the distribution of ecological effects and, in turn, evolutionary dynamics. Altogether these results show how fundamental properties of microbial growth are sufficient to generate complex ecological behavior, underscoring the necessity of considering ecology in studies of microbial evolution.

## Results

### Minimal model of microbial growth and competition over serial dilutions. We consider a microbial population consisting of

multiple strains with distinct growth traits, all competing for a single limiting resource. These strains may represent different microbial species, mutants of the same species, or even genetically-identical cells with purely phenotypic variation. We approximate the growth of each strain $i$ by the minimal model in Fig. 1a, defined by a lag time $\lambda_i$, exponential growth time $\tau_i$ (reciprocal growth rate, or time for the strain to grow $e$-fold), and yield $Y_i$, which is the population size supported per unit resource ('Methods')[33]. We assume resources are consumed in proportion to the total number of cells; it is straightforward to modify the model to other modes of resource consumption[21]. Therefore the amount of resources strain $i$ has consumed by time $t$ is $N_i(t)/Y_i$, where $N_i(t)$ is the population size of strain $i$. Growth stops when the amount of resources consumed by all strains equals the initial amount of resources; we define the initial density of resources per cell as $\rho$ ('Methods'). Although it is possible to consider additional growth traits such as a death rate or consumption of a secondary resource, here we focus on the minimal set of growth traits $\lambda_i$, $\tau_i$, and $Y_i$ since they are most often reported in microbial phenotyping experiments[23–29,34]. See Table 1 for a summary of all key notation.

The selection coefficient between a pair of strains $i$ and $j$ measures their relative ability to compete for resources[35,36]:

$$s_{ij} = \log\left(\frac{x_i'}{x_j'}\right) - \log\left(\frac{x_i}{x_j}\right), \qquad (1)$$

where $x_i$ is the density (dimensionless fraction of population size) of strain $i$ at the beginning of the competition and $x_i'$ is the density at the end. If new resources periodically become available, as occur in both laboratory evolution experiments and seasonal natural environments[33], then the population will undergo cycles of lag, growth, and saturation (Fig. 1b). Each round of competition begins with the same initial density of resources $\rho$. The population grows until all the resources are consumed, and then it is diluted down to the original size again; we assume the time to resource depletion is always shorter than the time between dilutions. We also assume the growth traits $\lambda_i$, $\tau_i$, and $Y_i$ of each strain remain the same over multiple competition rounds. The selection coefficients in Eq. 1 measure the rate of change of a strain's density $x_i$ over many rounds of these competitions ('Methods').

**Contribution of multiple growth traits to selection.** We can solve for the selection coefficients in Eq. 1 in terms of the strains' traits $\{\lambda_k, \tau_k, Y_k\}$, the initial strain densities $\{x_k\}$, and the initial density of resources per cell $\rho$ (Supplementary Note 1):

$$s_{ij} \approx s_{ij}^{\text{lag}} + s_{ij}^{\text{growth}} + \sum_k s_{ijk}^{\text{coupling}}, \qquad (2)$$

where

$$
\begin{aligned}
s_{ij}^{\text{lag}} &= -\frac{\bar{\tau}}{\tau_i \tau_j} \Delta\lambda_{ij}, \\
s_{ij}^{\text{growth}} &= -\frac{\bar{\tau}}{\tau_i \tau_j} \Delta\tau_{ij} \log(\rho\bar{Y}), \\
s_{ijk}^{\text{coupling}} &= -\frac{\bar{\tau}\bar{Y}}{\tau_i \tau_j} \frac{x_k}{\tau_k Y_k} \left( \Delta\tau_{ik}\Delta\lambda_{kj} - \Delta\lambda_{ik}\Delta\tau_{kj} \right).
\end{aligned}
\qquad (3)
$$

Here $\Delta\lambda_{ij} = \lambda_i - \lambda_j$ and $\Delta\tau_{ij} = \tau_i - \tau_j$ denote the pairwise differences in lag and growth times, while

$$\bar{\tau} = \frac{\sum_k \frac{x_k}{Y_k}}{\sum_k \frac{x_k}{\tau_k Y_k}}, \qquad \bar{Y} = \frac{1}{\sum_k \frac{x_k}{Y_k}} \qquad (4)$$

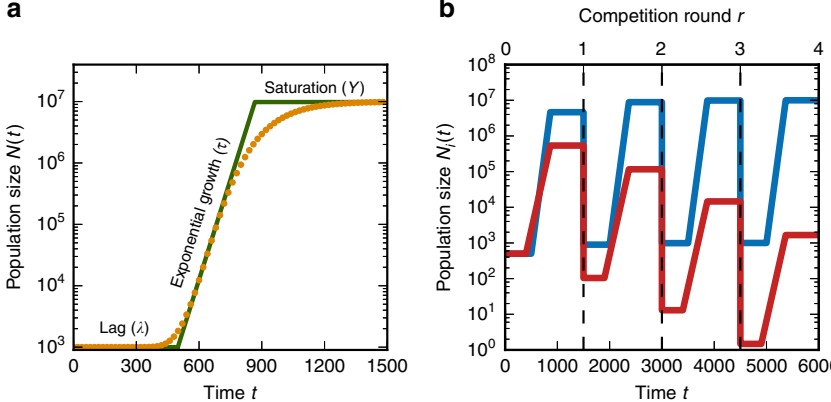

**Fig. 1** Model of growth and competition. **a** Approximation of a hypothetical growth curve (orange points) by the minimal three-phase model (green; see 'Methods'). Each phase is characterized by a quantitative trait: lag time $\lambda$, growth time $\tau$ (reciprocal growth rate), and yield $Y$ at saturation. **b** Growth curves of two competing strains over multiple rounds of competition in the model. Vertical dashed lines mark the beginning of each round, where the population is diluted down to the same initial population size with new resources ('Methods')

| Table 1 Summary of key notation | |
|---|---|
| **Definition** | **Notation** |
| Lag time of strain $i$ | $\lambda_i$ |
| Exponential growth time (reciprocal growth rate) of strain $i$ | $\tau_i$ |
| Yield (cells per resource) of strain $i$ | $Y_i$ |
| Density (fraction of population) of strain $i$ at beginning of competition round | $x_i$ |
| Density of resources per cell at beginning of competition round | $\rho$ |
| Effective exponential growth time of whole population (harmonic mean) | $\bar{\tau} = \dfrac{\sum_k \frac{x_k}{Y_k}}{\sum_k \frac{x_k}{Y_k \tau_k}}$ |
| Effective yield of whole population (harmonic mean) | $\bar{Y} = \dfrac{1}{\sum_k \frac{x_k}{Y_k}}$ |
| Lag-growth tradeoff | $c = -\left(\dfrac{\lambda_i - \lambda_j}{\tau_i - \tau_j}\right)$ |

are, respectively, the effective exponential growth time (reciprocal growth rate) and effective yield for the whole population (Supplementary Note 2). Since both of these quantities are harmonic means over the population, they are dominated by the smallest trait values. Therefore the effective growth time $\bar{\tau}$ for the whole population will generally be close to the growth time of the fastest-growing strain (smallest $\tau_k$), while the effective yield $\bar{Y}$ will generally be close to the yield of the least-efficient strain (smallest $Y_k$).

As Eq. 2 indicates, selection consists of three distinct additive components. The first is selection on the lag phase $s_{ij}^{\text{lag}}$, which is nonzero only if $i$ and $j$ have unequal lag times (Eq. 3). The second component is selection on the growth phase $s_{ij}^{\text{growth}}$, which is similarly nonzero only if $i$ and $j$ have unequal growth times. The relative magnitude of selection on growth versus lag is modulated by the density of resources $\rho$ and the effective population yield $\bar{Y}$:

$$\frac{s_{ij}^{\text{growth}}}{s_{ij}^{\text{lag}}} = \frac{\Delta \tau_{ij}}{\Delta \lambda_{ij}} \log(\rho \bar{Y}). \qquad (5)$$

In particular, increasing the resources $\rho$ leads to an increase in the magnitude of relative selection on growth versus lag, since it means the growth phase occupies a greater portion of the total competition time.

If $i$ and $j$ are the only two strains present, then the total selection on strain $i$ relative to $j$ is the net effect of selection on the lag and growth phases: $s_{ij} = s_{ij}^{\text{lag}} + s_{ij}^{\text{growth}}$ [21]. Figure 2a qualitatively shows this selection coefficient as a function of strain $i$'s lag and growth traits relative to those of strain $j$. If strain $i$'s traits fall in the blue region, the overall selection on it relative to strain $j$ will be positive, while if strain $i$'s traits fall in the red region, it will be negatively selected relative to strain $j$. Between these two regions lies a conditionally-neutral region (green), where strain $i$ will be positively selected at some densities and negatively selected at others [21]. The slope of the conditionally-neutral region is $\log(\rho \bar{Y})$ according to Eq. 5.

**Pairwise selection coefficients are modified by additional strains through higher-order effects**. If more than two distinct strains are present, then selection between $i$ and $j$ is modified by higher-order effects from the other strains. These higher-order effects are separate from the effects of increasing the initial population size upon addition of more strains, which simply decreases the initial density of resources $\rho$; we therefore hold $\rho$ constant (i.e., by scaling up the total amount of resources or scaling down the initial population size for each strain) when considering the addition of another strain. The higher-order modifications occur through three mechanisms, all fundamentally a consequence of having a finite resource. The first mechanism is through changes to the effective population growth time $\bar{\tau}$, which

 

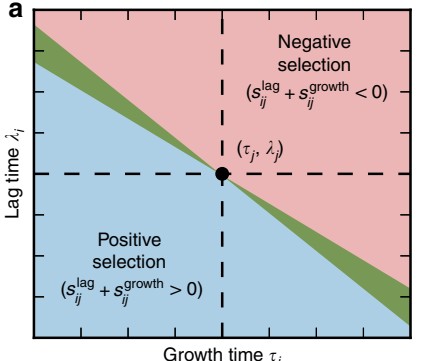
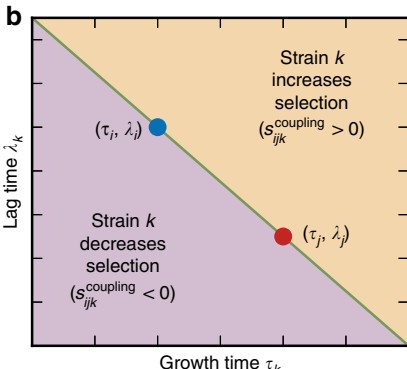

**Fig. 2** Selection in lag-growth trait space. **a** Diagram of selection on lag and growth for strain $i$ relative to strain $j$. Trait values of strain $j$ are marked by a black dot in the center. If the traits of strain $i$ lie in the blue region, $i$ is positively selected over strain $j$, while if strain $i$ lies in the red region, it is negatively selected. If strain $i$ lies in the green region, it is conditionally neutral with $j$ (positively selected at some densities and negatively selected at others). **b** Diagram of lag and growth times for strain $k$ relative to two other strains $i$ (blue) and $j$ (red). If the traits of strain $k$ lie in the orange region (above the straight line joining $i$ and $j$), then its coupling term $s_{ijk}^{\text{coupling}}$ (Eq. 3) increases the total selection coefficient of $i$ over $j$, while if $k$ lies in the violet region (below the straight line), then it decreases the selection of $i$ over $j$

rescales all selection coefficients (Eq. 3). For example, the addition of a strain with much faster growth will reduce the time all strains have to grow (Eq. 4), and thereby decrease the magnitude of all selection coefficients. The second modification is through the effective population yield $\bar{Y}$. Like $\bar{\tau}$, $\bar{Y}$ is a harmonic mean over strains, and similarly it will be significantly reduced if a strain with very low yield is added. This may change even the signs of some selection coefficients, since changes in $\bar{Y}$ modify the relative selection on growth versus lag between strains (Eq. 5).

Higher-order effects in $\bar{\tau}$ and $\bar{Y}$ are non-specific in the sense that these parameters are shared by all pairs of strains in the population. In contrast, the third type of modification is through the terms $s_{ijk}^{\text{coupling}}$, which couple the relative lag and growth traits of a pair $i$ and $j$ with a third strain $k$ (Eq. 3). This effect is specific, since each additional strain $k$ modifies the competition between $i$ and $j$ differently, depending on its growth traits and density $x_k$. We can interpret this effect graphically by considering the space of lag and growth times for strains $i$, $j$, and $k$ (Fig. 2b). If strain $k$ lies above the straight line connecting strains $i$ and $j$ in lag-growth trait space, then the coupling term will increase selection on whichever strain between $i$ and $j$ has faster growth (assumed to be strain $i$ in the figure). This is because strain $k$ has relatively slow growth or long lag compared to $i$ and $j$, thus using fewer resources than if the strains all had the same lag times or growth times. This then leaves more resources for $i$ and $j$, which effectively increases the selection on growth between the two strains beyond the $s_{ij}^{\text{growth}}$ term. If strain $k$ instead lies below the straight line, then it increases selection on the strain with slower growth, since $k$ uses more resources than if the strains all had the same lag times or growth times. For example, even if strain $i$ has both better growth and better lag compared to strain $j$, a third strain $k$ could actually reduce this advantage by having sufficiently short lag. Note that the coupling term is zero if all three strains have equal lag times or equal growth times. These coupling effects will furthermore be small if the relative differences in lag and growth traits are small, since $s_{ijk}^{\text{coupling}}$ is quadratic in $\Delta\lambda$ and $\Delta\tau$ while $s_{ij}^{\text{lag}}$ and $s_{ij}^{\text{growth}}$ are linear. In the following sections, we will demonstrate how these three higher-order mechanisms lead to nontrivial ecological dynamics.

**Growth tradeoffs enable neutral coexistence and multistability of many strains on a single resource**. Selection is frequency-dependent since $s_{ij}$ (Eqs. 2 and 3) depends on the densities $\{x_k\}$[21].

It is therefore possible for the population dynamics to have a fixed point ($s_{ij} = 0$ for all strains $i$ and $j$) at a nontrivial set of densities, giving rise to neutral coexistence or multistability (Supplementary Note 3). An unlimited number of distinct strains can have this property as long as they share a linear tradeoff between lag and growth times (Fig. 3a):

$$\lambda_i = -c\tau_i + \text{constant} \qquad (6)$$

for all $i$ and some parameter $c > 0$, which we define as the lag-growth tradeoff. The resource density $\rho$ must also fall in the range (Fig. 3b)

$$\frac{e^c}{\max_k Y_k} < \rho < \frac{e^c}{\min_k Y_k}. \qquad (7)$$

Note that $\rho > 1/\min_k Y_k$ is necessary as well, since if $\rho$ is below this limit there will be insufficient resources for some strains to grow at all. Since this limit is always lower than the upper bound in Eq. 7 (because $c > 0$), there will always be some range of $\rho$ at which the population has a fixed point.

Intuitively, a fixed point occurs because the strains consume resources in such a way to exactly balance selection on lag and growth for all pairs of strains. The linear lag-growth tradeoff across all strains from Eq. 6 causes the higher-order coupling terms $s_{ijk}^{\text{coupling}}$ of the selection coefficient to be zero (Eq. 3, Fig. 2b). It also means there is some value of the effective yield $\bar{Y}$ that will enable $s_{ij}^{\text{lag}} + s_{ij}^{\text{growth}} = 0$ for all pairs $i$ and $j$; this critical value of the effective yield is $\bar{Y} = e^c/\rho$ (Eq. 5, Supplementary Note 3). The constraint on resource density $\rho$ (Eq. 7) ensures that the population can actually achieve this required effective yield given the yield values of the individual strains.

These fixed points give rise to neutral coexistence, multistability, or a combination of both depending on the covariation between growth and yield across strains. The space of fixed-point densities is $(M - 2)$-dimensional if there are $M$ strains in the community satisfying the criteria in Eqs. 6 and 7 (Supplementary Note 3, Supplementary Fig. 1). Density fluctuations within this space are neutral, while fluctuations orthogonal to this space will be stable if there is also a tradeoff in growth and yield (Supplementary Note 3, Supplementary Fig. 2a), in addition to the lag-growth tradeoff (Eq. 6). In this case, an unlimited number of strains can neutrally coexist within this space of densities until

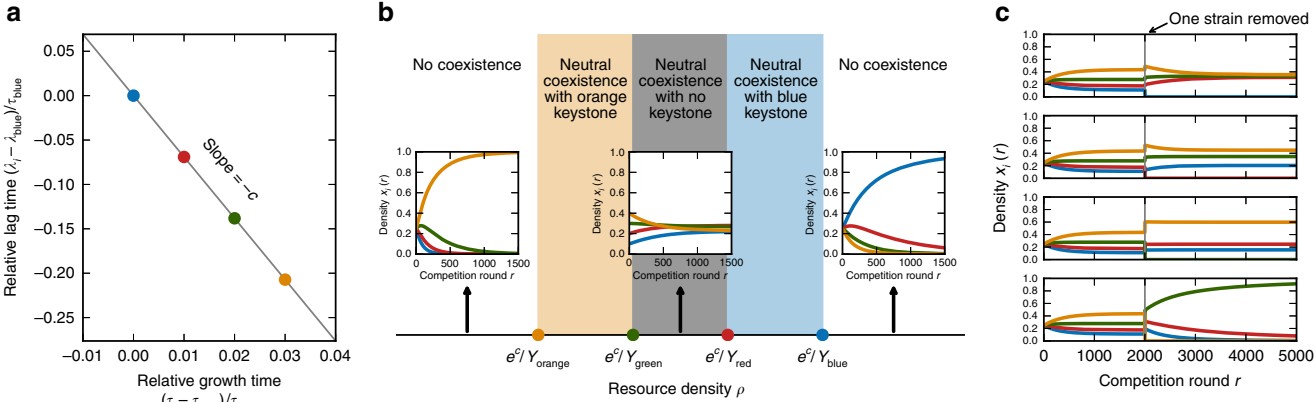

**Fig. 3** Neutral coexistence of multiple strains on a single resource. **a** Lag and growth times of four strains (blue, red, green, orange). For them to neutrally coexist in a community, these traits must have a linear tradeoff with slope $-c$ (Eq. 6). **b** Diagram of competition outcomes as a function of resource density $\rho$. Each inset shows the dynamics of the strains' densities $x_i(r)$ over rounds of competition $r$ for a particular value of $\rho$. All four strains will neutrally coexist if $\rho$ is in the range defined by Eq. 7 (shaded regions). If $\rho$ is in the orange or blue regions, coexistence hinges on a single keystone strain of corresponding color (orange or blue), while if $\rho$ is in the gray region, coexistence is robust to loss of any single strain. **c** Density dynamics of the same four strains with resource density $\rho$ in the orange region of **b**, so that the orange strain is the keystone. All four strains coexist together at first, then at competition round 2000 one strain is removed (different in each panel) and the remaining strains are allowed to reach their steady state. See Supplementary Note 8 for parameter values

genetic drift eventually leads to extinction of all but two strains. However, the time scale of this neutral coexistence will typically be very long compared to laboratory experiments or the time scales of other perturbations (new mutations or environmental changes), since the time scale of genetic drift (in units of competition rounds) is of order the bottleneck population size. While real strains will not exactly obey Eq. 6, even noisy tradeoffs can allow effective neutral coexistence over finite but significant time scales (Supplementary Note 3, Supplementary Fig. 3). If growth and yield have a synergy across strains rather than a tradeoff, the community will be multistable, dominated by different individual strains or pairs of strains depending on the initial conditions (Supplementary Fig. 2b, c).

**Neutral coexistence may hinge on a single keystone strain.** Besides small fluctuations in densities, an even stronger perturbation to a community is to remove one strain entirely. The stability of ecosystems in response to removal of a strain or species has long been an important problem in ecology; in particular, species whose removal leads to community collapse are known as keystone species due to their importance in stabilizing the community[31,32].

Neutrally-coexisting communities in our model will have a keystone strain for a certain range of resource density $\rho$. Figure 3b shows a diagram of competition outcomes across $\rho$ values for four hypothetical strains (blue, red, green, orange): if $\rho$ is in the orange or blue ranges, then removal of the strain of corresponding color (orange or blue) will cause rapid collapse of the community (all remaining strains but one will go extinct), since $\rho$ will no longer satisfy Eq. 7 for the remaining strains. Therefore the orange or blue strain is the keystone. However, if $\rho$ is within the gray region, then the community is robust to removal of any single strain. This shows that the keystone must always be the least-efficient or most-efficient strain (smallest or highest yield $Y_k$) in the community. Figure 3c shows the population dynamics with each strain removed from a coexisting community where the orange strain is the keystone.

Besides removal of an existing strain, another important perturbation to a community is invasion of a new strain, either

by migration or from a mutation. If the lag and growth times of the invader lie above the diagonal line formed by the coexisting strains' traits (e.g., as in Fig. 3a), then the invader will quickly go extinct (Supplementary Note 4). This would be true even if the invader has a growth time or lag time shorter than those of all the coexisting strains. On the other hand, if the invader lies below the diagonal line in lag-growth trait space, then it will either take over the population entirely or coexist with one of the original strains if it is sufficiently close to the diagonal line. It cannot coexist with more than one of the original strains, since all three points by assumption will not lie on a straight line in the lag-growth trait space.

**Pairwise competitions do not predict community behavior.** A fundamental issue for microbial ecology and evolution is whether pairwise competitions are sufficient to predict how a whole community will behave[5–8]. For example, if several strains coexist in pairs, will they coexist all together? Or if a single strain wins all pairwise competitions, will it also win a mixed competition with all strains present? We now show that competition for a single limiting resource with tradeoffs in growth traits is sufficient to confound these types of predictions due to the higher-order effects in the selection coefficient (Eqs. 2 and 3).

First, strains that coexist in pairs will generally not coexist all together. Strains $i$ and $j$ that coexist as a pair are characterized by a particular lag-growth tradeoff $c_{ij} = -\Delta\lambda_{ij}/\Delta\tau_{ij}$ (Eq. 6). For a set of these pairs to coexist all together, these tradeoffs must all be equal, which will generally not be the case. However, if the lag-growth tradeoffs are equal for all pairs, then the strains can indeed coexist in a community, but not at the same resource densities as for the pairs (Supplementary Note 5).

Second, in a collection of strains, a champion strain that wins all pairwise competitions may not prevail in a mixed competition of all strains. For example, in Fig. 4a the green strain beats the blue and orange strains one-on-one with a hoarding strategy— shorter lag with lower yield, but slower growth—but together the blue and orange strains consume resources efficiently enough to use their faster growth to beat green (Fig. 4b). In purely competitive models, this is a unique consequence of higher-order effects in the selection coefficients: the presence of the orange

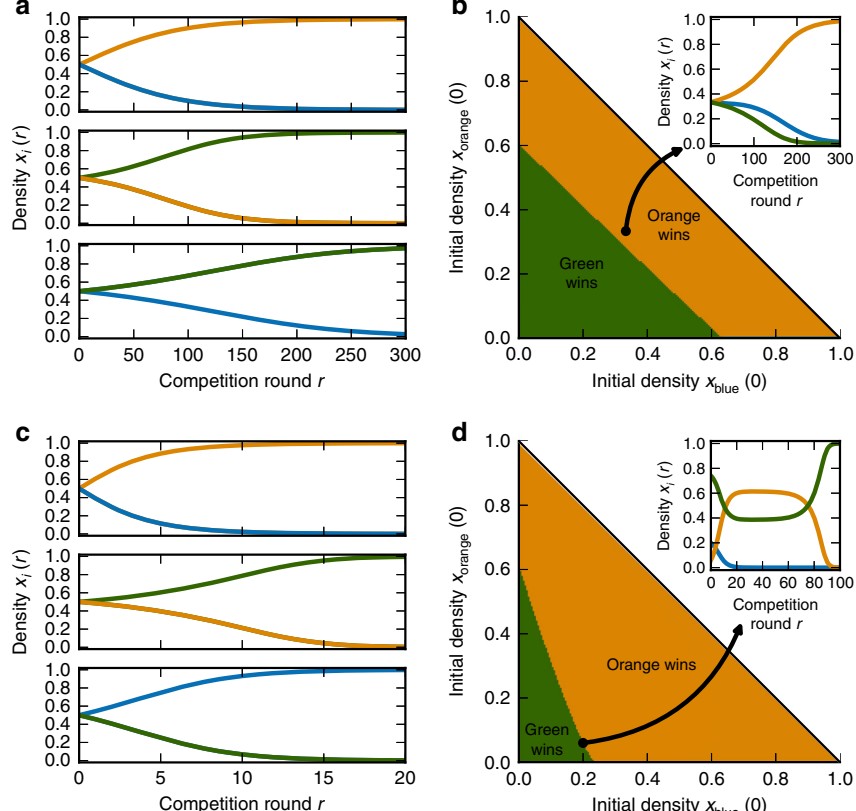

**Fig. 4** Pairwise competitions do not predict community behavior. **a**, **b** Example of three strains (blue, orange, green) with a single pairwise champion (green). Panel **a** shows density dynamics $x_i(r)$ for pairwise competitions, while **b** shows outcome of mixed competition as a function of initial conditions: orange marks space of initial densities where the orange strain eventually wins, while green marks initial densities where green eventually wins. Inset: density dynamics starting from equal initial densities (marked by black dot in main panel). **c**, **d** Same as **a**, **b**, but for three strains without a pairwise champion (non-transitivity). See Supplementary Note 8 for parameter values

strain actually changes the sign of the selection coefficient between green and blue (from positive to negative), and the blue strain similarly changes the sign of selection between green and orange. In this example it occurs via modifications to the effective population yield $\bar{Y}$. Even if the strains have identical yields, it is possible for the pairwise champion to lose the mixed competition over short time scales due to effects from the lag-growth coupling terms $s_{ijk}^{coupling}$ (Supplementary Note 6, Supplementary Fig. 4).

Third, it is also possible that there is no pairwise champion among a set of strains, meaning that selection is non-transitive[22]. For example, in Fig. 4c, orange beats blue and green beats orange, but blue beats green, forming a rock-paper-scissors game[37,38]. This outcome relies crucially on the existence of tradeoffs between growth traits, so that no single growth strategy always wins (Supplementary Note 7, Supplementary Fig. 5). In this example, orange beats blue by having a shorter lag time, green beats orange by growing faster and using resources more efficiently (higher yield), and blue beats green by having shorter lag and hoarding resources (lower yield). Non-transitivity in this model occurs only for pairwise competitions where each strain starts with equal density ($x_i(0) = 1/2$). Invasion competitions, where each strain competes against another starting from very low density (as would occur in an invasion by a migrant or a new mutant), cannot demonstrate this type of non-transitivity; however, invasions may not be simply transitive, either, if some pairs are bistable (Supplementary Note 7, Supplementary Fig. 5).

Non-transitive competitions are particularly confounding for predicting the behavior of a mixed community. Since there is no clear champion, non-transitive pairwise competitions are often hypothesized as the basis for oscillations or coexistence in mixed communities[22,37,38]. However, a non-transitive set of strains will not coexist all together in our model. Which strain wins, though, is not directly predictable from the pairwise selection coefficients, and in fact may depend on the initial conditions due to frequency-dependent selection. For example, Fig. 4d shows the outcomes of mixed competitions for a non-transitive set of strains as a function of their initial densities. If green starts at sufficiently high density, then it wins the mixed competition, but otherwise orange wins. In the inset we show one such mixed competition, with initial conditions on the boundary between the orange and green regimes. Here the outcome is very sensitive to the initial conditions, since frequency-dependent higher-order effects from the decaying blue population draw the orange and green strains toward their unstable fixed point, where they temporarily remain until the blue strain goes extinct and either orange or green eventually wins.

## Discussion

Variation in multiple growth traits is widespread in microbial populations[23–25], since even single mutations tend to be pleiotropic with respect to these traits[26,27]. Genetically-identical cells can also demonstrate significant growth variation[28,29]. We have

shown how this variation, with competition for only a single finite resource and no other interactions, is sufficient to produce a range of ecological phenomena, such as neutral coexistence, multistability, keystones, non-transitivity, and other collective behaviors where a community is more than the sum of its parts. This is because variation in multiple growth traits creates higher-order effects in which the pairwise selection coefficients themselves change in the presence of other strains. This goes beyond the effects of ordinary clonal interference[39]; for example, even the sign of the selection coefficients may change due to these higher-order effects, so that a strain that is the best in pairwise competitions actually goes extinct in the mixed community (Fig. 4a, b). For example, a mutation that is apparently beneficial against the wild-type alone may not only appear to be less beneficial in the presence of other mutations, but it could even appear to be deleterious. These results highlight the importance of considering the mutational distribution of ecological effects, rather than just fitness effects relative to a wild-type, for predicting evolutionary dynamics.

While previous work indicated that two strains may stably coexist through tradeoffs in growth traits[17–21], here we have shown that an unlimited number of strains can in fact coexist through this mechanism. Conceptually this is reminiscent of other coexistence mechanisms, such as the storage effect[40], where tradeoffs in multiple life-history traits allow long-term balancing of competition outcomes. A distinguishing feature of coexistence in our model is its neutrality, suggesting an additional mechanism by which neutrality may give rise to diversity[41]. Our work supports the hypothesis that higher-order effects should be widespread in microbial ecosystems[7,9]. Experimental tests for these effects and the predictive power of pairwise competitions remains limited, however. A recent study found that pairwise competitions of soil bacteria generally did predict the behavior of three or more species together[8], although there were important exceptions. Our results suggest an avenue for future investigations of this problem.

Coexistence and other key outcomes of the model require tradeoffs among lag, growth, and yield. The prevalence of these tradeoffs in microbial populations has been the subject of many previous studies, especially due to interest in the $r/K$ (growth-yield) selection problem. Some models of metabolic constraints do imply a tradeoff between growth and yield[42,43], while others propose that both tradeoffs and synergies are possible depending on the environment[44]; experiments have seen evidence of both cases[23–26].

The relationship between lag and growth has received less attention. While models of the lag phase suggest a synergy, rather than a tradeoff, with the growth phase ($c < 0$ in Eq. 6)[45–47], experimental support for this prediction has been mixed. For example, Ziv et al. found that in a large collection of yeast strains, faster growth mostly corresponded to shorter lag[29,48]. However, other sets of strains in yeast and E. coli have found no such trend[24,27]. Quantifying the prevalence and strength of these tradeoffs therefore remains an important topic for future investigation. Regardless of general trends, though, it is clear that lag-growth tradeoffs can be realized within some sets of microbial strains. For example, the tradeoff was directly observed in E. coli strains with certain mutations in adenylate kinase[27].

Given a collection of microbial strains and their measured growth traits, we can in principle use our model to predict the population dynamics of any combination of strains. If we also know the distribution of mutational effects on growth traits, we can further predict evolutionary dynamics to determine what patterns of traits are likely to evolve, which can be compared with experimental data[23–26]. In practice, real populations will likely contain more complex interactions beyond competition for a

single resource[19], as well as more complex growth dynamics[18]. Nevertheless, our model provides a valuable tool for interpreting the ecological and evolutionary significance of growth trait variation, especially for generating new hypotheses to be experimentally tested. For example, it can be used to estimate what role growth trait variation plays in the ecological dynamics of a coexisting community.

Our results are especially relevant to laboratory ecology and evolution experiments where populations undergo periodic growth cycles. While the importance of interference among mutants has been widely studied in these experiments[39,49], it is generally assumed that each mutant is described by a fixed selection coefficient independent of the background population, since the relative genetic homogeneity of the population suggests there should be no additional ecological interactions beyond competition for the limiting resource. But since even single mutations will produce variation in multiple growth traits, our results show that higher-order effects should actually be widespread in these populations. Even genetically-identical populations may experience higher-order effects due to stochastic cell-to-cell variation[28,29,45], although the effects will fluctuate from one round of competition to the next assuming cell-to-cell variation does not persist over these timescales. We look forward to quantifying the importance of these higher-order effects in future work.

## Methods

**Model of population growth and competition**. For a population consisting of a single microbial strain, we approximate its growth dynamics by the following minimal model (Fig. 1a)[50]:

$$N(t) = \begin{cases} N(0) & 0 \le t < \lambda, \\ N(0)e^{(t-\lambda)/\tau} & \lambda \le t < t_{\text{sat}}, \\ N(0)e^{(t_{\text{sat}}-\lambda)/\tau} & t \ge t_{\text{sat}}, \end{cases} \quad (8)$$

where $\lambda$ is the lag time during which no growth occurs and $\tau$ is the exponential growth time (reciprocal growth rate, or time over which the population grows $e$-fold). The saturation time $t_{\text{sat}}$ at which growth stops is determined by the amount of resources in the environment. We assume that the population size at saturation $N(t_{\text{sat}})$ is proportional to the total amount $R$ of the limiting resource. Let $Y$ denote this constant of proportionality ($N(t_{\text{sat}}) = RY$), which we will refer to as the intrinsic yield since it is the total number of cells per unit resource[33]. Let $\rho = R/N(0)$ be the initial density of resources per cell. The saturation time then equals

$$t_{\text{sat}} = \lambda + \tau \log(\rho Y). \quad (9)$$

If there are multiple distinct strains simultaneously competing for the same pool of resources, let each strain $i$ grow according to Eq. 8 with its own initial size $N_i(0)$ and growth traits $\lambda_i$, $\tau_i$, and $Y_i$. The initial density of each strain is therefore $x_i = N_i(0)/\sum_k N_k(0)$ and the initial density of resources is $\rho = R/\sum_k N_k(0)$. Since the total amount of resources used by strain $i$ by time $t$ is $N_i(t)/Y_i$, the saturation time $t_{\text{sat}}$ for the whole population is defined by

$$R = \sum_i \frac{N_i(t_{\text{sat}})}{Y_i}. \quad (10)$$

By solving this equation for $t_{\text{sat}}$ either numerically or analytically (Supplementary Note 1), we can calculate all properties of the competition, such as the densities of each strain at the end. While we have assumed here that resources are consumed in proportion to the total number of cells, which holds for resources such as space, it is straightforward to modify the model for other modes of resource consumption[21]. For example, resources may be consumed in proportion to the total number of cell divisions. The difference in these two models, however, will be negligible if the fold-change of the population over the growth cycle is large.

**Population dynamics over competition rounds**. If the population undergoes multiple rounds of dilution and resource renewal (Fig. 1b), the density of strain $i$ at the end of a round equals its density at the beginning of the next round (ignoring stochastic effects of sampling[21]). Let $x_i(r)$ be the density of strain $i$ at the beginning of competition round $r$ and $x'_i(r)$ be the density at the end, so that $x'_i(r) = x_i(r + 1)$. The selection coefficients determine how the densities change over the round. Using the selection coefficient definition $s_{ij} = \log\left(x'_i(r)/x'_j(r)\right) -$

$\log(x_i(r)/x_j(r))$ (Eq. 1), we can obtain the recurrence relation for the change in densities over each round:

$$x_i(r+1) = \frac{x_i(r)}{\sum_k x_k(r) e^{s_{ki}(\mathbf{x}(r))}}, \tag{11}$$

where $\mathbf{x}(r)$ is the vector of densities $\{x_k(r)\}$ at the beginning of round $r$. In all figures we calculate density trajectories by numerically solving the saturation equation (Eq. 10) for each competition round, and then iterating over rounds using Eq. 11. These dynamics, however, can also be approximated by a differential equation over a large number of rounds:

$$\begin{aligned}\frac{dx_i}{dr} &= \frac{x_i}{\sum_k x_k e^{s_{ki}(\mathbf{x})}} - x_i \\ &\approx x_i \sum_k x_k s_{ik}(\mathbf{x}),\end{aligned} \tag{12}$$

where on the second line we have invoked the approximation that all $s_{ki}$ values are small. This is of Lotka-Volterra form where the selection coefficients encode the effective (density-dependent) interaction coefficients between strains.

**Data availability**. Methods necessary to reproduce all analytical and numerical results are fully described in the article and Supplementary Information.

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

## Acknowledgements

We thank Tal Einav for his detailed comments on the manuscript. This work was supported by NIH awards F32 GM116217 to M.M. and R01 GM068670 to E.I.S.

## Author contributions

M.M. and E.I.S. designed research; M.M. performed calculations; M.M. wrote the manuscript. Both authors edited and approved the final version.

## Additional information

**Competing interests:** The authors declare no competing interests.

