## [Peer Review File · Nature Communications]

Reviewers' comments:

Reviewer #1 (Remarks to the Author):

In this manuscript, Manhart and Shakhnovich employ a simple analytical model to study the ecology and evolution of multi-species microbial communities competing for a single limiting resource. In previous work, the authors have used this model to study pairwise interactions, and show that, in an environment experiencing seasonal fluctuations, the presence of a lag time can allow pairs to coexist, create dependencies on initial conditions, and form non-transitive interactions. Here, they extend this work to multispecies communities. They find that the presence of additional species genetically modifies pairwise interactions, forming higher-order interactions. These higher-order interactions confound prediction based on pairwise interactions, and allow multiple species to coexist if there are trade-off in their trait values.

This find this work to be highly novel, interesting, and well executed. Higher-order interactions are the focus of a lot of attention recently, as they are recognized as a mechanism that can potentially be central to understanding the structure and function of species-rich microbial communities. Nonetheless, there is still little work that provides intuition into how such interaction arise in simple model, and what are their implications. Thus, this simple, analytically tractable model provides a significant contribution to the field, and will be of interest to the microbiome research community, as well as to ecologists in general. My comments, listed below, are mainly regarding clarity and presentation. I also note a few additions that can add value to the manuscript, but I do not feel these are required for publication.

Specific comments

I found the description of the model in the main text hard to parse. I would have benefited from a more detailed exposition in the main text, making the author's modeling choices more explicit. For example, that the resource consumption is proportional to cell number rather than divisions, and that the time between dilutions is always longer than the time to resource depletion.

Having the resource density be the same at the beginning of each cycle (Lines 112-115) implies that the dilution factor, or amount of added resource can vary across cycles, as the community composition and total density at the end of cycles changes. This is not the typical experimental setup, nor is it likely to occur in natural settings. How important is this modeling choice? Would results be significantly altered if we imposed a fixed dilution factor into a fixed resource concentration, as is typical in experiments?

The selection coefficient between pairs is affected by the resource concentration, as well as the presence of additional species. The effect of additional species is mediated by their resource consumption, so it would be very illuminating to directly compare the effect of adding a species, to that of lowering the resource by the amount consumed by that species.

The precise effect of adding an additional species on the selection coefficient is not quite clear to me. That is, how does the overall selection coefficient for a pair, rather than just the coupling part, change due to the presence of an additional species. At the moment, the effect is distributed between the non-specific effect present in all terms of eq. 2. Intuitively, an additional species can only decrease the growth time, thus favoring species with a shorter lag time. Relatedly, I find the phrasing "using fewer resources than expected" (lines 200-202) confusing. Do you mean fewer than the population average?

I find the language regarding coexistence of multiple species to be overly strong to the point of being misleading. First, I would not consider an unstable fixed point to be coexistence, and the title of section C: "Growth tradeoffs produce stable communities of multiple strains" is certainly not supported by that section. Next, coexistence of multiple (or infinite) species in this model requires fine tuning of parameters with precise tradeoffs. Fig. S2 is certainly interesting, and shows that imperfect trade-offs can produce long species-rich transients, but it does not show what is the typical number of species that exhibit long-term coexistence in this model. I believe such analysis will be very valuable.

The language about application to experiments is overly bold. As noted by the authors, experiments are likely to display much richer dynamics than the model, and involves complex interactions, including nutrient recycling and toxin secretion. Thus, it remains to be seen how much predictive power this model actually offers.

I found the use of the term "without direct interactions" in the title to be vague. Many would consider resource competition to be a direct interaction, and reserve the term "indirect interaction" to describe interactions mediated by changes in abundances of additional species (as in ref. 8). What would the authors consider a direct interaction in the context of their model? A more informative title can explicitly mention higher-order interactions, or single limiting resources.

The discussion mentions previous works on coexistence mechanism (lines 409-413), but fails to reference works by Chesson on the storage effect (e.g. Chesson, Peter, and Nancy Huntly. "The roles of harsh and fluctuating conditions in the dynamics of ecological communities." *The American Naturalist* 150.5 (1997): 519-553.). This mechanism seems to me to be especially relevant to this model, and elucidating the connection between them will help ecologists fit it into existing established frameworks.

Line 326-327: Actually, the generalized Lotka-Volterra system, which includes only pairwise interactions, can produce similar outcomes. That is, a species that beats either of two competitors in pairwise competitions, but is excluded in the three-way competition.

Lines 364-366: Non-transitivity in this model requires at least one of the pairs to exhibit multi-stability (since invasions are transitive). Therefore, it is not clear whether the dependence on initial conditions and difficulty in predicting outcomes discussed here are indeed a consequence of the non-transitivity or of that multi-stability.

In the example of Fig. 4A,B is there bistability in the green-red pair? If not, the trio outcomes of red winning is not a stable point.

The word density is used to refer to dimensionless fractions, rather than # cells/volume (line 106, supplement section S2). This should be clarified in the main text.

Line 433: Should say 'agreed with these predictions' or 'confirmed these predictions'.

Reviewer #2 (Remarks to the Author):

This is a well written and interesting paper, dealing with an important open question in microbial ecology: How do variations in growth traits affect ecological interactions? The authors use an elegant model to show that high order effects arising from variation in growth lag and yield can lead to complex multi-species communities when there is a trade-off in these growth traits and the only interaction is competition for a single resource. Moreover, they find that communities that emerge due

to these effects are not predicted from selection coefficients obtained in pairwise competitions, both in terms of presence-absence at equilibrium and abundance at equilibrium. This ties in with an important ongoing debate concerning the predictability of microbial community assembly based on low-dimensional information such as pairwise competition coefficients and growth rates in monocultures. Overall, I believe that with some straightforward revisions the paper will be ready for acceptance at Nature Communications.

Main Comments:

1. The paper is attempting to address two different sets of ideas simultaneously. The first deals with the concept of fitness (measured as selection rate) in experimental evolution, and at its core this paper is showing that high-order ecological effects can radically change the magnitude and direction of fitness effects. The second is one of community ecology and conditions for coexistence. The introduction and final section of the conclusion as written currently address the former, whereas most of the results and discussion deals with the latter. Moreover the community ecology part of the paper is where most of its originality comes from. I would recommend that the authors shift gears and present this in the introduction as an ecology paper about higher-order interactions rather than an evolution paper about intransitive fitness.

2. If the authors placed their results in the context of empirical papers it would broaden the paper's appeal as well as substantiate some of the assumptions made in the paper. For example one of the main assumptions of section C is that of a strict linear trade-off between growth and lag. Whilst the supplement does cover this further, by demonstrating that even a noisy trade-off can delay extinction if a trade-off is strict enough, it's unclear how strict the trade-off is likely to be amongst natural strains and therefore how generally applicable the results in this section are. The authors could easily address this by briefly placing their results in the context of published experimental measurements of the lag-growth trade-off.

Reviewer #3 (Remarks to the Author):

The authors provide a comprehensive theoretical analysis of a frequently encountered situation where multiple strains compete for a single resource in a well-mixed, serial-dilution (feast and famine) environment and differ only in their lag times, growth rates and yields. The main claim is that if there is a tradeoff between growth rate and lag times (so that strains that grow better have longer lag times and vice versa) complex stable communities are possible. Moreover, collections of strains can exhibit counter-intuitive phenomena: e.g. the winner of all pairwise competitions does not win for some initial concentrations when all the strains are mixed.

My major concern is that, despite the careful analysis, the authors are misinterpreting the significance of their findings. They do not seem to demonstrate a tradeoff leading to complex community formation. Rather, the communities of more than two strains that they describe constitute cases of neutral coexistence. It is generic and unsurprising that in a multi-dimensional phenotypic space you can find a set of measure zero (in this case a straight line), where strains will exhibit neutral coexistence. The only truly stable communities they describe are the two-strain ones, and as the authors note, it has been known for a long time (both theoretically and experimentally) that two strains can stably coexist in this way. This neutral coexistence is evident from the analysis in supplementary section S6 and Fig. S3. The authors find that all but one eigenvalue is zero. The communities will drift along the purple line of Fig. S3a until only two strains remain. The authors do not use the term "stable coexistence" in a way that is consistent with the ecological literature. To summarize, what the authors show is that if you can tune the properties of strains with infinite

precision, you can design collections of strains that can neutrally coexist, which is well-known and unsurprising.

This is not to say that near-neutral coexistence is not important, particularly in short-term microcosm experiments. In evolutionary setting, this has been known as clonal interference. The careful analysis and nice analytical results that the authors provide can be very useful for interpreting results of serial-dilution microcosm experiments. I would suggest reframing all the results as an exploration of the conditions for near neutral coexistence in the practically important setting of serial-dilution, well-mixed microbial experiments.

We appreciate the positive responses and constructive suggestions from all three reviewers. Please find below detailed point-by-point replies (following their original comments in italics) and descriptions of our corresponding revisions. All significant changes to the manuscript are highlighted in red.

Reviewer 1

In this manuscript, Manhart and Shakhnovich employ a simple analytical model to study the ecology and evolution of multi-species microbial communities competing for a single limiting resource. In previous work, the authors have used this model to study pairwise interactions, and show that, in an environment experiencing seasonal fluctuations, the presence of a lag time can allow pairs to coexist, create dependencies on initial conditions, and form non transitive interactions. Here, they extend this work to multi-species communities. They find that the presence of additional species genetically modifies pairwise interactions, forming higher-order interactions. These higher-order interactions confound prediction based on pairwise interactions, and allow multiple species to coexist if there are trade-off in their trait values.

This find this work to be highly novel, interesting, and well executed. Higher-order interactions are the focus of a lot of attention recently, as they are recognized as a mechanism that can potentially be central to understanding the structure and function of species-rich microbial communities. Nonetheless, there is still little work that provides intuition into how such interaction arise in simple model, and what are their implications. Thus, this simple, analytically tractable model provides a significant contribution to the field, and will be of interest to the microbiome research community, as well as to ecologists in general. My comments, listed below, are mainly regarding clarity and presentation. I also note a few additions that can add value to the manuscript, but I do not feel these are required for publication.

We thank the reviewer for his/her interest in our work.

Specific comments

I found the description of the model in the main text hard to parse. I would have benefited from a more detailed exposition in the main text, making the author's modeling choices more explicit. For example, that the resource consumption is proportional to cell number rather than divisions, and that the time between dilutions is always longer than the time to resource depletion.

As suggested by the reviewer, we have clarified key details of the model in the main text (lines 92–96, 121–122). We have also added a complete Methods section at the end of the main text (in accordance with journal style), which should make all the model details more readily accessible.

Having the resource density be the same at the beginning of each cycle (Lines 112–115) implies that the dilution factor, or amount of added resource can vary across cycles, as the community composition and total density at the end of cycles changes. This is not the typical experimental setup, nor is it likely to occur in natural settings. How important is this modeling choice? Would results be significantly altered if we imposed a fixed dilution factor into a fixed resource concentration, as is typical in experiments?

In this work we considered a fixed bottleneck size (with varying dilution factor) for simplicity, since it results in a constant effective population size. However, we agree that the case of fixed dilution factor is also very important. We are currently preparing a separate work which will fully explain this case, but briefly, the selection coefficient result (Eq. 2), including the higher-order effects emphasized in this paper, holds for any single competition round regardless of the dilution scheme. The dynamics over multiple competition rounds (Eq. 10), however, must be supplemented by the dynamics of the initial resource density ρ , which will change from round to round as the bottleneck population size fluctuates. Coexistence can still occur, although the conditions on parameters and densities are somewhat different.

The selection coefficient between pairs is affected by the resource concentration, as well as the presence of additional species. The effect of additional species is mediated by their resource consumption, so it would be very illuminating to directly compare the effect of adding a species, to that of lowering the resource by the amount consumed by that species.

The precise effect of adding an additional species on the selection coefficient is not quite clear to me. That is, how does the overall selection coefficient for a pair, rather than just the coupling part, change due to the presence of an additional species. At the moment, the effect is distributed between the nonspecific effect present in all terms of eq. 2. Intuitively, an additional species can only decrease the growth time, thus favoring species with a shorter lag time. Relatedly, I find the phrasing “using fewer resources than expected” (lines 200–202) confusing. Do you mean fewer than the population average?

We agree that our explanation of the effect on selection between strains i and j of adding strain k was unclear. Certainly adding strain k to an existing population of i and j will trivially change selection between i and j simply by reducing the amount of resources available to i and j — in terms of our model, the initial density of resources ρ will decrease. Therefore, when we compare selection on strains i and j in the context of a binary competition between i and j alone, versus in a population with strain k also present, we keep ρ fixed. This means that upon addition of strain k , either the total amount of resources is increased, or the supplemented population is diluted down to the original size. Otherwise, the change in selection between i and j will be a conflation of the “intrinsic” higher-order effects that are our focus here with the effects of just reducing resource density. We have clarified this issue in the text on lines 176–183.

Our phrase “using fewer resources than expected” was intended to mean fewer resources than expected if the strains were neutral, i.e., having the same growth and lag traits. We have clarified this in the text on lines 214–215 and 220–221.

I find the language regarding coexistence of multiple species to be overly strong to the point of being misleading. First, I would not consider an unstable fixed point to be coexistence, and the title of section C: “Growth tradeoffs produce stable communities of multiple strains” is certainly not supported by that section. Next, coexistence of multiple (or infinite) species in this model requires fine tuning of parameters with precise tradeoffs. Fig. S2 is certainly

interesting, and shows that imperfect tradeoffs can produce long species-rich transients, but it does not show what is the typical number of species that exhibit long-term coexistence in this model. I believe such analysis will be very valuable.

We have revised the section on coexistence to be more clear about its properties, especially its neutrality and multistability (lines 232–285). Regarding the number of species that can exhibit long-term coexistence, we agree this would be interesting analysis, but since strains are defined by points in a continuous space of growth traits, any results will depend significantly on the underlying distribution of traits from which strains are drawn. We therefore believe this would require a more detailed study of what that distribution should be (e.g., based on first principles or empirical estimates), which would go beyond the scope of the present paper.

The language about application to experiments is overly bold. As noted by the authors, experiments are likely to display much richer dynamics than the model, and involves complex interactions, including nutrient recycling and toxin secretion. Thus, it remains to be seen how much predictive power this model actually offers.

We have revised the description of application to experiments on lines 488–504 to address the reviewer’s concerns.

I found the use of the term “without direct interactions” in the title to be vague. Many would consider resource competition to be a direct interaction, and reserve the term “indirect interaction” to describe interactions mediated by changes in abundances of additional species (as in ref. 8). What would the authors consider a direct interaction in the context of their model? A more informative title can explicitly mention higher-order interactions, or single limiting resources.

We agree that the phrase “without direct interactions” in the title is potentially misleading. As suggested, we have replaced this with “on a single limiting resource.”

*The discussion mentions previous works on coexistence mechanism (lines 409–413), but fails to reference works by Chesson on the storage effect (e.g. Chesson, Peter, and Nancy Huntly. “The roles of harsh and fluctuating conditions in the dynamics of ecological communities.” *The American Naturalist* 150.5 (1997): 519–553.). This mechanism seem to me to be especially relevant to this model, and elucidating the connection between them will help ecologists fit it into existing established frameworks.*

We thank the reviewer for referring us to the storage effect example. We have added a reference to this on lines 447–451.

Line 326–327: Actually, the generalized Lotka-Volterra system, which includes only pairwise interactions, can produce similar outcomes. That is, a species that beats either of two competitors in pairwise competitions, but is excluded in the three-way competition.

We thank the reviewer for pointing out the possibility of the generalized Lotka-Volterra model allowing the pairwise champion to lose the ternary competition. We intended to restrict ourselves to competitive models that can be defined by selection coefficients. We have clarified this in the text (lines 360–361).

Lines 364–366: Non-transitivity in this model requires at least one of the pairs to exhibit multi stability (since invasions are transitive). Therefore, it is not clear whether the dependence on initial conditions and difficulty in predicting outcomes discussed here are indeed a consequence of the non-transitivity or of that multi-stability.

We agree that our discussion of predicting community dynamics from pairwise competitions conflated the effects of non-transitivity with frequency-dependent effects (multistability). The strong dependence on the initial conditions is fundamentally a consequence of the frequency-dependent selection, since in principle non-transitive pairwise competitions can produce a ternary outcome that is independent of initial conditions (while still being difficult to predict from the pairwise competitions alone). We have clarified this in the text (lines 401 and 409).

In the example of Fig. 4A,B is there bistability in the green-red pair? If not, the trio outcomes of red winning is not a stable point.

There is indeed bistability in the red-green pair of Fig. 4A,B (red has faster growth and higher yield but longer lag): green only beats red when it is above density ≈ 0.32 . This is why green dominates the ternary competition as well if it starts at sufficiently high density.

The word density is used to refer to dimensionless fractions, rather than # cells/volume (line 106, supplement section S2). This should be clarified in the main text.

We have clarified our definition of density at the beginning of the Results section (lines 111–112) and in Table I.

Line 433: Should say “agreed with these predictions” or “confirmed these predictions”.

We have revised the text including the typo formerly on line 433.

Reviewer 2

This is a well written and interesting paper, dealing with an important open question in microbial ecology: How do variations in growth traits effect ecological interactions? The authors use an elegant model to show that high order effects arising from variation in growth lag and yield can lead to complex multispecies communities when there is a tradeoff in these growth traits and the only interaction is competition for a single resource. Moreover, they find that communities that emerge due to these effects are not predicted from selection coefficients obtained in pairwise competitions, both in terms of presence/absence at equilibrium and and

abundance at equilibrium. This ties in with an important ongoing debate concerning the predictability of microbial community assembly based on lowdimensional information such as pairwise competition coefficients and growth rates in monocultures. Overall, I believe that with some straightforward revisions the paper will be ready for acceptance at Nature Communications.

We thank the reviewer for his/her interest in our work and support for publication of our manuscript.

Main Comments:

1. The paper is attempting to address two different sets of ideas simultaneously. The first deals with the concept of fitness (measured as selection rate) in experimental evolution, and at it's core this paper is showing that high-order ecological effects can radically change the magnitude and direction of fitness effects. The second is one of community ecology and conditions for coexistence. The introduction and final section of the conclusion as written currently address the former, whereas most of the results and discussion deals with the later. Moreover the community ecology part of the papers is where most of it's originality comes from. I would recommend that the authors shift gears and present this in the introduction as an ecology paper about higher-order interactions rather than an evolution paper about intransitive fitness.

We agree that the Introduction's attempt to address both the distribution of fitness effects and the importance of higher-order interactions was unclear. We have therefore removed the material about fitness effects from the Abstract and Introduction (lines 7–25) and strengthened the focus on ecological effects of higher-order interactions.

2. If the authors placed there results in the context of empirical papers it would broaden the papers appeal as well as substantiate some of the assumptions made in the paper. For example one of the main assumptions of section C is that of a strict linear trade-off between growth and lag. Whilst the supplement does cover this further, by demonstrating that even a noisy trade-off can delay extinction if a trade-off is strict enough, it?s unclear how strict the trade-off is likely to be amongst natural strains and therefore how generally applicable the results in this section are. The authors could easily address this by briefly placing their results in the context of published experimental measurements of the lag-growth trade-off.

We agree that a more detailed discussion of existing data on growth-lag tradeoffs is valuable. We have added this to the Discussion on lines 472–486.

Reviewer 3

The authors provide a comprehensive theoretical analysis of a frequently encountered situation where multiple strains compete for a single resource in a well-mixed, serial-dilution (feast and famine) environment and differ only in their lag times, growth rates and yields. The main claim is that if there is a tradeoff between growth rate and lag times (so that strains that grow better have longer lag times and vice versa) complex stable communities are possible.

Moreover, collections of strains can exhibit counter-intuitive phenomena: e.g. the winner of all pairwise competitions does not win for some initial concentrations when all the strains are mixed.

My major concern is that, despite the careful analysis, the authors are misinterpreting the significance of their findings. They do not seem to demonstrate a tradeoff leading to complex community formation. Rather, the communities of more than two strains that they describe constitute cases of neutral coexistence. It is generic and unsurprising that in a multi dimensional phenotypic space you can find a set of measure zero (in this case a straight line), where strains will exhibit neutral coexistence. The only truly stable communities they describe are the two-strain ones, and as the authors note, it has been known for a long time (both theoretically and experimentally) that two strains can stably coexist in this way. This neutral coexistence is evident from the analysis in supplementary section S6 and Fig. S3. the authors find that all but one eigenvalue is zero. The communities will drift along the purple line of Fig. S3a until only two strains remain. The authors do not use the term “stable coexistence” in a way that is consistent with the ecological literature. To summarize, what the authors show is that if you can tune the properties of strains with infinite precision, you can design collections of strains that can neutrally coexist, which is well-known and unsurprising.

This is not to say that near-neutral coexistence is not important, particularly in short-term microcosm experiments. In evolutionary setting, this has been known as clonal interference. The careful analysis and nice analytical results that the authors provide can be very useful for interpreting results of serial-dilution microcosm experiments. I would suggest reframing all the results as an exploration of the conditions for near neutral coexistence in the practically important setting of serial-dilution, well-mixed microbial experiments.

We agree that we did not clearly explain the nature of coexistence in our model, especially its neutral and multistable aspects. We have significantly revised this section to more accurately explain these details (lines 232–285), as well as updating our terminology throughout the Abstract, Introduction, and Discussion. While it is true that only two strains can coexist under truly stabilizing selection in our model, as the reviewer points out, a coexisting community with a larger number of strains may persist for a significant period of time since the neutral dynamics will typically be slow. Therefore we believe this coexistence mechanism may be biologically relevant in some cases, especially in laboratory evolution experiments as the reviewer suggests. We have further emphasized the application of our results to these experiments in the Discussion (lines 488–504).

REVIEWERS' COMMENTS:

Reviewer #1 (Remarks to the Author):

I thank the authors for their careful revision of the manuscript. They have satisfactorily addressed all of my comments, and I am happy to recommend this manuscript for publication.

Reviewer #2 (Remarks to the Author):

The authors have adequately addressed my comments and I am satisfied with their modifications to the paper.

Reviewer #3 (Remarks to the Author):

The authors did a much better job emphasizing that the coexistence described in this manuscript is neutral coexistence rather than the default stable coexistence. There are still several places where the two notions are (implicitly) confused.

On lines 442-447 they say: "The ability to coexist on a single limiting resource contradicts the principle of competitive exclusion [15, 16]. While previous work indicated that two strains may stably coexist through tradeoffs in growth traits [17, 21] here we have shown that an unlimited number of strains can in fact coexist through this mechanism."

It is only the ability to stably coexist on a single limiting resource that contradicts the principle of competitive exclusion, and the previous work they cite is also about stable coexistence. Therefore, to say that "unlimited number of strains can coexist through this mechanism" is misleading in this context, since stable coexistence is implied.

Lines 232/233, the section title: "coexistence" should be replaced by "neutral coexistence"

It is important to emphasize that neutral coexistence does not only apply to section 234 – 285 but throughout. Therefore:

Line 286 section title should be changed to: "Neutral coexistence may hinge on a single keystone strain." This not only clarifies the finding but also makes it more interesting. It is well known that coexistence relying on stabilizing interactions can hinge on a keystone species, but the fact that neutral coexistence can hinge on it is more unexpected. It emphasizes that ecological neutrality is not the same as ecological equivalence. It elaborates on the point the authors make that neutrality is not clonal interference.

Fig. 3 title: Coexistence should be changed to ◊ Neutral coexistence

This problem is manifested in the title as well. "... complex microbial communities on a single limiting resource" would strongly suggest to most ecologists that a mechanism is proposed for overcoming competitive exclusion, i.e. a mechanism leading to stable coexistence of many strains. (The wording of the title is evocative because the principle of competitive exclusion is normally phrased as "no two species can stably coexist on the same limiting resource"). Neutral or near neutral coexistence has never been a problem on a single limiting resource.

The word “produce” in the title is also a bit misleading. In the paper, no process is described that spontaneously generates communities in which many strains neutrally coexist. It is only demonstrated that such communities can be constructed. A more accurate title is: “Growth tradeoffs allow long-persisting communities on a single limiting resource”.

The importance of neutral processes for diversity has been well appreciated (e.g. “Unified neutral theory of biodiversity”). Some of that literature can be referenced.

Multi-stability is only mentioned in the main text. If it is a main finding that makes it to the abstract, it deserves at least a stand-alone paragraph and maybe figure panel. Moreover, Lines 237/238 imply that Supplementary Methods Sec. S3, Fig. S1 discuss multi-stability. They do not. There is no section on multi-stability in Suppl. It is only implicit in Fig. S3. What are the conditions for multi-stability? Is it only possible for collections of strains that can neutrally coexist?

We appreciate the positive responses from all three reviewers once again. Please find below detailed point-by-point replies (following their original comments in italics) and descriptions of our corresponding revisions. All significant changes to the main text are highlighted in red.

Reviewer 1

I thank the authors for their careful revision of the manuscript. They have satisfactorily addressed all of my comments, and I am happy to recommend this manuscript for publication.

We appreciate the reviewer's positive response and support for publication.

Reviewer 2

The authors have adequately addressed my comments and I am satisfied with their modifications to the paper.

We appreciate the reviewer's positive response and support for publication.

Reviewer 3

The authors did a much better job emphasizing that the coexistence described in this manuscript is neutral coexistence rather than the default stable coexistence. There are still several places where the two notions are (implicitly) confused.

We appreciate the reviewer's positive response to our revisions, and thank him/her for the careful consideration of our coexistence results.

On lines 442–447 they say: “The ability to coexist on a single limiting resource contradicts the principle of competitive exclusion [15, 16]. While previous work indicated that two strains may stably coexist through tradeoffs in growth traits [17, 21] here we have shown that an unlimited number of strains can in fact coexist through this mechanism.”

It is only the ability to stably coexist on a single limiting resource that contradicts the principle of competitive exclusion, and the previous work they cite is also about stable coexistence. Therefore, to say that “unlimited number of strains can coexist through this mechanism” is misleading in this context, since stable coexistence is implied.

We agree the comparison to the principle of competition exclusion is misleading; we have therefore removed this reference.

Lines 232/233, the section title: “coexistence” should be replaced by “neutral coexistence” It is important to emphasize that neutral coexistence does not only apply to section 234–285 but throughout. Therefore:

Line 286 section title should be changed to: “Neutral coexistence may hinge on a single keystone strain.” This not only clarifies the finding but also makes it more interesting. It is well known that coexistence relying on stabilizing interactions can hinge on a keystone species, but the fact that neutral coexistence can hinge on it is more unexpected. It emphasizes that

ecological neutrality is not the same as ecological equivalence. It elaborates on the point the authors make that neutrality is not clonal interference.

Fig. 3 title: Coexistence should be changed to Neutral coexistence

We have implemented these three changes as recommended (lines 231 and 291 and Fig. 3 caption). We have also more systematically clarified that coexistence is neutral throughout the main text (e.g., lines 301, 431, 457–460) and Supplementary Information (e.g., Supplementary Note 3 and Supplementary Figs. 2 and 3).

This problem is manifested in the title as well. "... complex microbial communities on a single limiting resource" would strongly suggest to most ecologists that a mechanism is proposed for overcoming competitive exclusion, i.e. a mechanism leading to stable coexistence of many strains. (The wording of the title is evocative because the principle of competitive exclusion is normally phrased as "no two species can stably coexist on the same limiting resource"). Neutral or near neutral coexistence has never been a problem on a single limiting resource.

The word "produce" in the title is also a bit misleading. In the paper, no process is described that spontaneously generates communities in which many strains neutrally coexist. It is only demonstrated that such communities can be constructed. A more accurate title is: "Growth tradeoffs allow long-persisting communities on a single limiting resource".

We appreciate the reviewer's concerns about the title, but we would strongly prefer to maintain the current title. We believe "produce" is appropriate because the paper indeed shows that growth tradeoffs produce the various phenomena described in the paper; it is true that the paper does not discuss mechanisms for producing those growth tradeoffs in the first place, but the title does not claim this. We would prefer to use the language "complex communities" over "long-persisting communities" because the former more accurately includes the several interesting phenomena that arise from growth tradeoffs and higher-order interactions discussed in the paper, of which neutral coexistence is just one example.

The importance of neutral processes for diversity has been well appreciated (e.g. "Unified neutral theory of biodiversity"). Some of that literature can be referenced.

We thank the reviewer for suggesting this connection; we have added it to the Discussion (lines 457–460).

Multistability is only mentioned in the main text. If it is a main finding that makes it to the abstract, it deserves at least a standalone paragraph and maybe figure panel. Moreover, Lines 237/238 imply that Supplementary Methods Sec. S3, Fig. S1 discuss multistability. They do not. There is no section on multistability in Suppl. It is only implicit in Fig. S3. What are the conditions for multistability? Is it only possible for collections of strains that can neutrally coexist?

We agree that our discussion of multistability was not fully clear, especially as it relates to neutral coexistence. We have significantly revised this section of the main text (lines 233–290) and Supplementary Information (Supplementary Note 3) to discuss both more clearly. They were conflated because they are actually two cases of the same more fundamental issue, which is the existence of nontrivial fixed points. Equations 6 and 7 and Supplementary Note 3 define the conditions under which these fixed points exist. These fixed points correspond to neutral coexistence if growth and yield have a tradeoff, whereas they give rise to multistability if growth and yield have a synergy. We have more clearly illustrated the relationship between these two cases in Supplementary Note 3 and the caption to Supplementary Fig. 2.